# Persistent Local Homology in Graph Learning

**Minghua Wang**                                                          *minghuaw@buffalo.edu*
*Department of Computer Science and Engineering*
*State University of New York at Buffalo, NY, USA*

**Yan Hu**                                                               *yan.hu@kaust.edu.sa*
*Computer, Electrical and Mathematical Sciences and Engineering Division*
*King Abdullah University of Science and Technology, Saudi Arabia*

**Ziyun Huang**                                                           *zxh201@psu.edu*
*Department of Computer Science and Software Engineering*
*Penn State Erie, The Behrend College, PA, USA*

**Di Wang**                                                              *di.wang@kaust.edu.sa*
*Computer, Electrical and Mathematical Sciences and Engineering Division*
*King Abdullah University of Science and Technology, Saudi Arabia*

**Jinhui Xu**                                                            *jinhui@buffalo.edu*
*Department of Computer Science and Engineering*
*State University of New York at Buffalo, NY, USA*

**Reviewed on OpenReview:** *https://openreview.net/forum?id=qunyX9WYr6*

## Abstract

In this study, we introduce *Persistent Local Homology* (PLH) for graphs, a novel method that synergizes persistent homology with local homology to analyze graph structures. We begin by mathematically formalizing PLH, defining it as the application of persistent homology to annular local subgraphs. This foundation paves the way for the development of a computational pipeline, specifically tailored for PLH, which we explore in various graph learning contexts. Despite its utility, a complexity analysis reveals potential computational bottlenecks in PLH application. To address this, we propose *Reduced PLH* (rPLH), an efficient variant designed to significantly lower computational complexity. Experimental evaluations with rPLH demonstrate its capability to retain the effectiveness of the original PLH while substantially reducing computational demands. The practical utility of PLH and rPLH is further corroborated through comprehensive experiments on both synthetic and real-world datasets, highlighting their broad applicability and potential in diverse analytical scenarios.

## 1 Introduction

Topological Data Analysis (TDA) has emerged as a powerful tool for understanding complex data structures, offering insights into the underlying geometry and topology of datasets. One of the key techniques in TDA is Persistent Homology (PH), which quantifies topological features across multiple scales and has been utilized in numerous tasks, proving its efficacy in capturing the topological properties of data (Edelsbrunner et al., 2002). The application of PH has notably extended to graph-based tasks, highlighting its growing importance in understanding graph structures and properties.

In the realm of graph learning, PH provides global topological signatures that lead to successful topology-driven machine learning approaches for graphs (Carrière et al., 2020; Hofer et al., 2020; Chen et al., 2021a; Hofer et al., 2017; 2019; Horn et al., 2022; Rieck et al., 2019; Wong & Vong, 2021). The evolution of PH, particularly through variants like Zigzag persistence and multipersistence, has been instrumental in

advancing spatial-temporal graph neural network tasks, offering refined analysis tools for these dynamic systems (Chen et al., 2021b; 2022). Moreover, the field has seen innovative integrations of PH into neural networks, where learnable filtration and vectorization methods have been utilized for end-to-end learning (Carrière et al., 2020; Hofer et al., 2020; Leygonie et al., 2021). This integration has substantially enriched the predictive capabilities of graph models, demonstrating the versatility and effectiveness of topological methods in complex graph-based learning scenarios.

Building upon these advancements, the application of PH on local subgraphs has recently become a focal point in graph learning. This shift towards a more localized approach represents a significant evolution in the use of PH, aiming to tackle unique challenges within subgraph structures. For example, Zhao et al. (2020) proposed a network architecture that integrates PH in local subgraphs to optimize message passing in graph convolution processes. Similarly, Chen et al. (2021a) utilized PH to analyze node neighborhoods, incorporating topological summaries to counter challenges like oversmoothing and vulnerability to graph perturbations. In link prediction tasks, Yan et al. (2021) leveraged PH on local subgraphs to analyze node interactions, enhancing the task's performance significantly.

**Motivation** Despite these successes, applying PH to local subgraph analysis has its limitations, primarily in potentially overlooking crucial local topological features. This issue is highlighted in Subsection 4.1, where we demonstrate that PH's application to local subgraphs might be insensitive to variations among these structures, which could limit its effectiveness in certain scenarios. To address this, our paper introduces the concept of Persistent Local Homology (PLH) in graphs. It aims to provide a more nuanced understanding of the topological characteristics of local graph structures, thereby overcoming the limitations of traditional PH in local subgraph analysis. This research presents several key questions, including the formalization of PLH on graphs, its impact on graph learning tasks, the computational complexities involved, and strategies for enhancing scalability. Our work is dedicated to exploring these questions, with the goal of broadening the understanding and application of PLH in graph learning.

**Contributions** We established the formulation of PLH on graphs, designed to reveal topological characteristics frequently omitted in traditional PH. Our novel method harnesses PLH's capacity for probing local structures, thereby enhancing both algorithms and applications in graph-based learning. To our knowledge, this is the first study to establish a linkage between PLH and graph learning paradigms. In addition, we propose an innovative variant, termed Reduced PLH (rPLH)[1], which retains comparable efficacy to PLH but with significantly reduced complexity. Our experimental results demonstrate that PLH not only consistently surpasses PH in synthetic datasets, but also significantly contributes to performance improvements in various real-world graph learning models. Furthermore, PLH has been shown to advance the state-of-the-art (SOTA) in graph learning models.

## 2 Related Works

**Persistent Homology Variants** PH has been a powerful tool for data analysis since its introduction by Edelsbrunner et al. (2000). It has been applied to a wide range of problems, including the study of protein structure, image analysis, and machine learning. In recent years, a number of variants of PH have been proposed. Extended persistence broadens the scope of persistence analysis by considering not only the sublevel sets of a real-valued function but also the superlevel sets (Cohen-Steiner et al., 2009). Zigzag persistent homology addresses the limitations of the original framework in dealing with dynamic data sets by introducing a more flexible construction of persistence modules (Carlsson et al., 2009). Multiparameter persistent homology extends the persistence concept to higher dimensions, enabling a more comprehensive understanding of data sets with complex structures (Carlsson & Zomorodian, 2007). Our work, PLH on graphs, can also be seen as a variant of PH on graphs since it extends the concept of PH to graphs by considering the topological properties on annular local subgraphs.

**Persistent Local Homology** The exploration of local homology in data analysis has been a subject of substantial interest in recent research. Bendich et al. (2007) and Bendich (2008) laid the groundwork by

---

[1]The term "reduced" in rPLH does not refer to reduced homology.

developing methods to assess local homology in stratified spaces and extending persistent homology theory to local homology, respectively. This enabled a deeper understanding of the local topological structures in data sets. Building on this, Dey et al. (2014) focused on estimating manifold dimensions using local homology derived from point samples. In a novel application, Ahmed et al. (2014) introduced a new distance metric for road network comparison, leveraging local persistent homology to simultaneously consider spatial proximity and local topology. Further, Fasy & Wang (2016) provided a survey of local homology's application in data analysis, emphasizing its efficacy in extracting local structures from datasets. Contrasting these approaches, Stolz et al. (2020) proposed an approximation of local homology through the absolute homology of small annuli within specified neighborhoods. In a similar vein, Von Rohrscheidt & Rieck (2023) introduced a multi-scale persistent framework for local homology, enhancing the capture of local geometric information.

**PLH in A Metric Space**   A pivotal work closely related to our study is that of Von Rohrscheidt & Rieck (2023). This research is grounded in a novel formulation of PLH in a metric space, which is instrumental in evaluating the shape of neighborhoods across multiple scales. A key application of PLH in their work is to locally estimate the intrinsic dimension of points within a given space. However, a fundamental distinction between our approach and that of Von Rohrscheidt & Rieck (2023) lies in a significant divergence in methodological requirements. Specifically, our approach does not necessitate the embedding of graphs within a metric space, thereby allowing for a broader application in scenarios where such embeddings are either impractical or infeasible. This distinction underscores the unique contribution of our work in the realm of graph analysis, setting it apart from the existing literature on PLH in metric spaces.

## 3   Preliminaries

To enhance the introduction of our methodology, we offer a succinct overview of the fundamental concepts integral to our approach. For readers seeking more in-depth technical insights, we recommend consulting the seminal works of Edelsbrunner & Harer (2022), Dey & Wang (2022), and Hatcher (2002), which provide comprehensive treatments of these topics.

### 3.1   Homology

Homology, a fundamental topological invariant, distinguishes between different topological spaces, offering a computationally feasible yet less stringent equivalence relation than homotopy. This paper mainly uses simplicial homology, investigating sequences of homology groups associated with a simplicial complex.

**Simplicial Complex**   A simplicial complex, denoted as $K$, is defined as a collection of simplices that adhere to two conditions: every face of a simplex in $K$ must also be in $K$, and the intersection of any two simplices in $K$ should be either a mutual face or empty. Simplices, generalizing entities like triangles or tetrahedra, can extend into higher-dimensional spaces.

**Simplicial Homology**   For a given simplicial complex $K$, the $p$-th chain group, $C_p(K)$, is formed as either a vector space or a free abelian group over the integers $\mathbb{Z}$. The generators of this group are the $p$-simplices of $K$. The introduction of boundary homomorphisms, $\partial_p : C_p(K) \to C_{p-1}(K)$, facilitates the definition of homology groups, defined as $H_p(K) = \text{Ker}(\partial_p)/\text{Im}(\partial_{p+1})$. These groups encapsulate critical topological information, with elements in $\text{Ker}(\partial_p)$ representing cycles, and those in $\text{Im}(\partial_{p+1})$ being identified as boundaries. The elements of the homology group $H_p(K)$, known as homology classes, are precisely those cycles that are not boundaries and represent the $p$-dimensional holes in the complex. They highlight the fundamental distinction between cycles and boundaries within homology. This distinction is central to the application of homology in algebraic topology (Hatcher, 2002).

### 3.2   Persistent Homology

**Persistent Homology**   Persistent homology formalizes the notion of topological simplification through a process called a filtration (Edelsbrunner et al., 2000). In this context, a filtration is defined as a nested sequence of simplicial complexes $\mathcal{F} = \{K_0, K_1, \ldots, K_m\}$, where each complex $K_i$ is a subcomplex of its

successor $K_{i+1}$, and $K$ in $\emptyset = K_0 \subseteq K_1 \subseteq \cdots \subseteq K_m = K$ represents the final simplicial complex in the sequence.

An important aspect of persistent homology is the mapping of homology groups through this sequence. The inclusion map from a complex $K_i$ to a subsequent complex $K_j$ $(i \leq j)$ induces a homomorphism $\iota_p^{i,j} : H_p(K_i) \to H_p(K_j)$, connecting the functorial nature of homology in topological data analysis. This results in a sequence of homology groups for a fixed dimension $p$, given by $0 = H_p(K_0) \to H_p(K_1) \to \ldots \to H_p(K_m) = H_p(K)$.

The $p$-th persistent homology groups, denoted by $H_p^{i,j}$, are the images of these induced homomorphisms for $0 \leq i \leq j \leq m$. They capture the homology classes of $K_i$ that persist through to $K_j$. The ranks of these groups, known as the $p$-th persistent Betti numbers, are expressed as $\beta_p^{i,j} = \mathrm{rank}\, H_p^{i,j}$.

Moreover, the count of $p$-dimensional homology classes that are born in the simplicial complex $K_i$ and cease to exist before reaching $K_j$ is defined as $\mu_p^{i,j}$, calculated using the formula: $\mu_p^{i,j} = (\beta_p^{i,j-1} - \beta_p^{i,j}) - (\beta_p^{i-1,j-1} - \beta_p^{i-1,j})$ for all $i < j$ and for each dimension $p$. This count reflects the birth and death of homology classes across the filtration.

**Persistence Diagram**  The Persistence Diagram (PD) for each dimension $p$, denoted as $\mathrm{dgm}_p(\mathcal{F})$, is constructed by plotting each pair $(i, j)$ with a multiplicity equal to $\mu_p^{i,j}$. These diagrams, essentially multisets, provide a comprehensive visual representation of the persistence of homology classes throughout the filtration. According to the fundamental lemma of persistent homology, these diagrams encode the entirety of the persistent homology groups' information.

**Bottleneck Distance**  Among the metrics used for comparing persistence diagrams, the bottleneck distance is particularly prevalent (Cohen-Steiner et al., 2005). For two persistence diagrams $\mathrm{dgm}_p$ and $\mathrm{dgm}_p'$, the bottleneck distance $d_B$ is defined as: $d_B(\mathrm{dgm}_p, \mathrm{dgm}_p') = \inf_\eta \sup_x \|x - \eta(x)\|_\infty$, where $x \in \mathrm{dgm}_p$ and $\eta$ denotes a bijection between the points of the diagrams. This definition incorporates points from the diagonal of the diagrams for correspondence when $\mathrm{dgm}_p$ and $\mathrm{dgm}_p'$ have differing numbers of points, facilitating a meaningful comparison between diagrams.

## 4  Methodology

In this section, we establish our methodology, centered around three key elements. First, we define the PLH in graphs, exploring its stability properties. Next, we detail the computation pipeline for PLH on graphs, including a comprehensive analysis of its computational complexity. Finally, we introduce a reduced method designed to lower this complexity, ensuring efficiency in large-scale applications.

### 4.1  Formalization of PLH on Graphs

**Local Subgraph and Annular Local Subgraph**  In graph theory, particularly concerning an undirected graph $G = (V, E)$, we introduce the notions of *local subgraph* $g^s(v)$ and *annular local subgraph* $g_r^s(v)$ to understand localized graph structures. For a vertex $v \in V$, the local subgraph $g^s(v)$ encompasses vertices within a distance $s$, defined as $\{u \in V \mid d(u, v) \leq s\}$, where $d(u, v)$ represents the shortest path in terms of hops. Extending this concept, the annular local subgraph $g_r^s(v)$ is obtained by excluding a smaller local subgraph $g^r(v)$ from $g^s(v)$, i.e., consisting of vertices $\{u \in V \mid r < d(u, v) \leq s\}$, offering a nuanced perspective by focusing on a ring-shaped region around $v$. As a special case, the *punctured local subgraph* $g_0^s(v)$ is obtained by excluding the central vertex $v$ from $g^s(v)$, thus providing insights into the vertex's immediate neighborhood.

**Local Homology on Graphs**  We employ the homology on annular local subgraphs to represent local homology within graphs. This approach necessitates a preliminary review of local homology as it pertains to topological spaces and simplicial complexes. In topological spaces, the local homology at a point $x$, denoted by $H_p(X, X \setminus \{x\})$, is conceptualized as a categorical colimit of the homology groups $H_p(X, X \setminus U)$ over neighborhoods $U$ of $x$ (Munkres, 2018; Hatcher, 2002; Skraba & Wang, 2014). This concept becomes

practically significant when interpreting $X$ as a simplicial complex, with $x$ as a constituent vertex. In this context, two pivotal constructs are introduced: the *Star of x*, denoted $\text{St}(x)$, which is the aggregate of simplices incorporating $x$, and the *Link of x*, denoted $\text{Lk}(x)$, comprising simplices in $\text{St}(x)$ but not including $x$ itself. Von Rohrscheidt & Rieck (2023) elucidated that the relative homology $H_p(X, X \setminus \{x\})$ is isomorphic to the reduced homology $\tilde{H}_{p-1}(\text{Lk}(x))$, by employing the excision axiom and exact sequences in reduced homology. This finding underscores the representational efficacy of $\text{Lk}(x)$ in capturing local homological characteristics at $x$. Since a graph can be viewed as a 1-dimensional simplicial complex, we formalize local homology in graphs as the homology on the punctured local subgraphs within 1 hop. Inspired by the approach in Von Rohrscheidt & Rieck (2023), we expand the punctured local subgraphs into annular local subgraphs and treat the inner and outer hops as hyperparameters to accommodate various requirements. Therefore, we formalize local homology on graphs to homology on annular local subgraphs.

**Topological Difference between Local Subgraph and Annular Local Subgraph** While at first glance, the distinction between a local subgraph and an annular local subgraph may appear subtle, the exclusion of the inner subgraph in the latter fundamentally alters its topological structure. This modification is not merely a reduction in the number of vertices and edges, but a transformation that can significantly impact the graph's intrinsic properties. To elucidate this concept, we present an example demonstrating how the removal of the inner subgraph in forming an annular local subgraph leads to a substantial change in the subgraph's topology. This example underscores the importance of considering such alterations when analyzing graph-based data structures, particularly in applications where topological characteristics play a pivotal role.

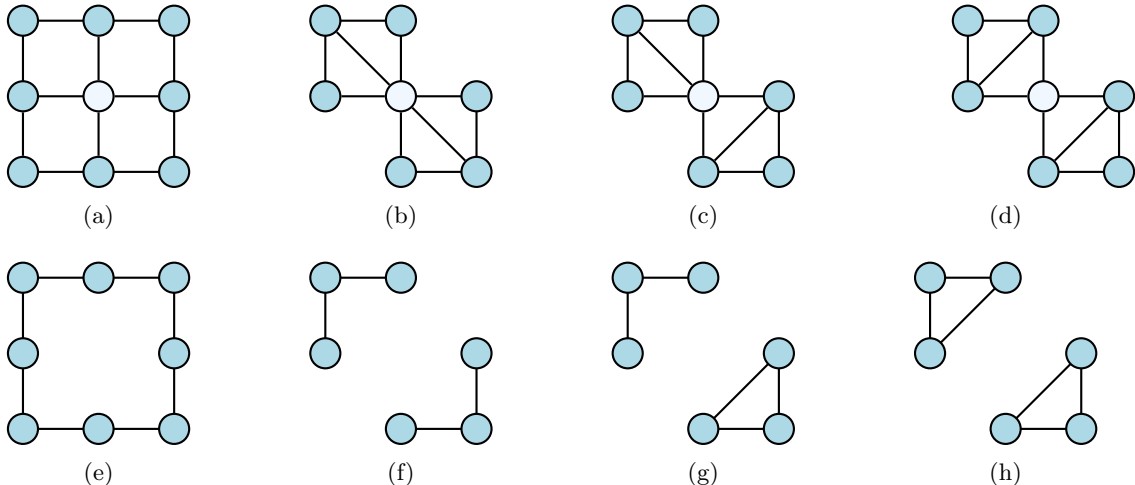

Figure 1: The upper and bottom rows illustrate local subgraphs and their corresponding annular local subgraphs $g_0^2$, respectively.

Table 1: Homologies of the subgraphs in Figure 1.

|  | Homology | (a) | (b) | (c) | (d) |
|---|---|---|---|---|---|
| Local subgraphs | $H_0$ | $\mathbb{Z}$ | $\mathbb{Z}$ | $\mathbb{Z}$ | $\mathbb{Z}$ |
|  | $H_1$ | $\mathbb{Z}^4$ | $\mathbb{Z}^4$ | $\mathbb{Z}^4$ | $\mathbb{Z}^4$ |
|  | Homology | (e) | (f) | (g) | (h) |
| Annular local subgraphs | $H_0$ | $\mathbb{Z}$ | $\mathbb{Z}^2$ | $\mathbb{Z}^2$ | $\mathbb{Z}^2$ |
|  | $H_1$ | $\mathbb{Z}$ | $0$ | $\mathbb{Z}$ | $\mathbb{Z}^2$ |

Given four distinct local subgraphs, as illustrated in the upper row of Figure 1, we derive their corresponding annular (or punctured) local subgraphs, shown in the bottom row, by excluding the central vertex from each.

To quantify and compare their topological differences, we treat these subgraphs as simplicial complexes and compute their homologies over $\mathbb{Z}$. The computed homologies are tabulated in Table 1, which presents the rank 0 and 1 homologies for each of the subgraphs depicted in Figure 1. An analysis of these homologies reveals a key insight: while the homologies of the upper row subgraphs (local subgraphs) fail to offer a distinct differentiation among them, a combination of $H_0$ and $H_1$ homologies of the bottom row subgraphs (annular local subgraphs) successfully distinguishes all four. This observation emphasizes the superior efficacy of homology applied to annular local subgraphs, which is adept at discerning topological subtleties that often elude detection through the homology of conventional local subgraphs. This example effectively demonstrates the added value of employing local homology on graphs for discerning subtle topological variations.

**Persistent Local Homology on Graphs** Persistent local homology, akin to persistent homology, is a method for discerning local topological features of a space across various spatial resolutions. This concept is defined as the persistent homology computed on an annular local subgraph, facilitating the detection of topological features in the vicinity of a central vertex. In essence, persistent local homology on graphs aims to delineate the persistent homology of annular local subgraphs, thereby uncovering the graph's persistent topological traits on a localized scale.

## 4.2 PLH Computation and Its Stability

**PLH Computation** Recall that our principal objective is to examine the efficacy of PLH in graph learning, a process that typically necessitates inputs in vector form. We define the operator $\mathbf{plh}_p : V \rightarrow \mathbb{R}^d$ as the mapping that transforms vertices into vectors. Consequently, $\mathbf{plh}_p(v)$ represents the vectorized $p$-th PLH for vertex $v$. In this paper, the operator is conceptualized as a composition of three functions:

$$\mathbf{plh}_p(v) = (\mathrm{PI} \circ \mathrm{dgm}_p \circ g_r^s)\,(v, \mathcal{F}),$$

with $v$ being the vertex under observation and $\mathcal{F}$ as the selected filtration function. Herein, $g_r^s$ produces the required annular local subgraph. The $\mathrm{dgm}_p$ function computes the $p$-th persistent diagram using matrix reduction or other advanced algorithms, yielding a finite multiset of two-dimensional points whose coordinates specify birth and death times for topological features. Subsequently, the persistence image PI converts a diagram into a finite-dimensional vector representation in $\mathbb{R}^d$ (Adams et al., 2017). For further details, we direct readers to the relevant surveys (Otter et al., 2017; Ali et al., 2022), as these topics possess their own distinct interests. Ultimately, by concatenating different $p$-th PLHs, we achieve the complete topological descriptor: $\mathbf{plh}(v) = \bigoplus_{p=0,\ldots,n} \mathbf{plh}_p(v)$ for each $v \in V$. This final outcome $\mathbf{plh}$ can be readily incorporated into various learning models, underscoring the versatility of our proposed approach.

The PLH computation hinges on the choice of filtration function, which is crucial for determining the topological descriptors. A common choice for filtration is the Vietoris-Rips filtration (Carlsson, 2009). This is also our main choice since the distances between pairs of vertices in a graph can be calculated from the shortest path based on the edge weighs or hops.

**Vietoris-Rips Filtration on An Annular Local Subgraph** Given a weighted graph $g_r^s = (\mathcal{V}, \mathcal{E}, w)$, where $\mathcal{V}$ is the set of vertices, $\mathcal{E}$ is the set of edges, and $w : \mathcal{E} \rightarrow \mathbb{R}_{\geq 0}$ is the weight function assigning non-negative real numbers as weights to the edges, the Vietoris-Rips filtration with respect to the shortest path distances is defined as a sequence of simplicial complexes, $\{\mathrm{VR}_\epsilon(g_r^s)\}_{\epsilon \geq 0}$, constructed as follows. For a non-negative real number $\epsilon$, the simplicial complex $\mathrm{VR}_\epsilon(g_r^s)$ is defined as:

- *Vertices*: Each vertex in $\mathcal{V}$ corresponds to a 0-simplex in $\mathrm{VR}_\epsilon(g_r^s)$.

- *Edges*: For each pair of vertices $u, v \in \mathcal{V}$, an edge (also as a 1-simplex) $\{u, v\}$ is included in $\mathrm{VR}_\epsilon(g_r^s)$ if and only if $d(u, v) \leq \epsilon$, where $d(u, v)$ is the shortest path distance between $u$ and $v$.

- *Higher-Dimensional Simplices*: Similarly, a set of vertices $\{v_0, v_1, \ldots, v_k\}$ forms a $k$-simplex in $\mathrm{VR}_\epsilon(g_r^s)$ if and only if all pairwise distances between these vertices are less than or equal to $\epsilon$. This means that for every pair of vertices $v_i, v_j$ in the set, $d(v_i, v_j) \leq \epsilon$.

The Vietoris-Rips filtration of $g_r^s$ is the collection $\{\text{VR}_\epsilon(g_r^s)\}_{\epsilon \geq 0}$, where each $\text{VR}_\epsilon(g_r^s)$ is a simplicial complex as defined above. As $\epsilon$ increases, more simplices are added to the complexes, capturing the topology of the graph at different scales.

**Stability of PLH**   The stability of the topological features in question is maintained with respect to perturbations in the edge weights of the annular local subgraph. This stability characteristic aligns with the previously established stability of the PI, as proven by Adams et al. (2017). To ensure the stability of PLH, we present a stability theorem specific to our context.

**Theorem 4.1.** *Let $g_r^s = (\mathcal{V}, \mathcal{E}, w)$ and $g_r^{s\prime} = (\mathcal{V}, \mathcal{E}, w')$ be two finite weighted subgraphs on the same vertex and edge sets but with different edge weight functions, we use Vietoris-Rips filtration w.r.t. the shortest path distance, i.e., $f(\epsilon) = \text{VR}_\epsilon(g_r^s)$ and $f'(\epsilon) = \text{VR}_\epsilon(g_r^{s\prime})$. Then, the bottleneck distance between their persistence diagrams $\text{dgm}_p$ and $\text{dgm}_p'$ for any dimension $p$ of homology is bounded by the maximum change in edge weights: $d_B(\text{dgm}_p, \text{dgm}_p') \leq \max_{e \in \mathcal{E}} \mid w(e) - w'(e) \mid$.*

### 4.3   Reduced PLH

Reduced PLH (rPLH) is designed as an alternative to the conventional PLH, offering enhanced scalability compared to the original PLH. In this section, we will analyze the complexity of PLH, subsequently introducing rPLH, and then examine its complexity.

**Complexity of PLH**   Persistent diagram can be computed in matrix multiplication time, for instance, in $\mathcal{O}(n^\omega)$, where $n$ denotes the number of simplices and $\omega = 2.376$ using Coppersmith-Winograd (Milosavljević et al., 2011), or $\omega = 2.372$ with more recent results (Duan et al., 2022). Considering a graph as a 1-dimensional simplicial complex, $n$ can be defined as $n = |E| + |V|$. In the context of a dense graph, $n$ subsequently becomes $\mathcal{O}(|V|^2)$, leading to the PD complexity of $\mathcal{O}(|V|^{2\omega})$. A critical observation in dense graphs is that they often cause dense local subgraphs. If we proceed to compute PD for each node, the worst-case complexity elevates to $\mathcal{O}(|V|^{2\omega+1})$. This has been empirically corroborated by our experiments, where we observed substantial slowness, hindering the practical applications of PLH extensively.

**Construction of Reduced Annular Local Subgraph**   To address this computational bottleneck, we propose reduced PLH. The main idea is to lower the complexity by restricting the total number of simplices, which can archive by restricting the total vertices in the annular local subgraph. To archive this, we introduce a maximum degree threshold, denoted as $D$. The vertices of a reduced annular local subgraph is given by Algorithm 1.

---

**Algorithm 1** Reduced annular local subgraph vertex set building algorithm.

---

1: $V_0 \leftarrow \{v\}$          ▷ Initialize with the given vertex $v$
2: **for** $i \leftarrow 1$ to $s$ **do**          ▷ Iterate over $s$ layers
3:      $V_i \leftarrow \emptyset$
4:      **for all** $u \in V_{i-1}$ **do**
5:          $V_i \leftarrow V_i \cup$ top-$D$ $i$-th edge layer vertices with highest degree connected to vertex $u$
6: **return** $V_{r+1} \cup \ldots \cup V_s$      ▷ Output the vertex set that induces annular local subgraph $g_r^s$

---

This algorithm essentially builds successive layers of vertices, with each layer being determined by the connectivity and degree of vertices in relation to the previous layer. The aim is to expand from a single initial vertex $v$ outward, layer by layer, selecting at each step the most connected vertices from the immediate neighborhood of the last layer. The threshold $D$ essentially set an upper bound for the total number of the connections from the vertex to the next layer. We can prioritize neighbors with higher degrees first, allowing the local subgraph to retain its topological structure to the highest feasible extent. We name the PH on the reduced annular local subgraph as the reduced PLH (rPLH). It is noteworthy that rPLH maintains the same stability as outlined in Theorem 4.1, provided that the reduced annular local subgraph is constructed.

**Complexity of rPLH**  The complexity of PD in rPLH for a graph is $\mathcal{O}(|V|D^{2s\omega})$, where $s$ is the maximum hop in a local subgraph. Notably, the complexity of rPLH can be significantly lower than PLH if we set $D^s \ll |V|$, a condition that we can fully control in practice. Furthermore, our experimental results demonstrate that a properly configured rPLH can achieve a level of effectiveness comparable to that of the standard PLH while tremendously reducing computation time. This proposed method offers a promising avenue for the efficient application of PLH in the context of complex graph structures.

### 4.4  PLH in Graph Learning

To further investigate the effectiveness of PLH in graph learning, we establish the required methods to construct the models used in the subsequent experiments. This section is divided into three parts. Initially, we introduce the key component in graph learning: GNNs. Then, we explore the integration of PLH into graph learning models. Finally, we present a model that establishes a new SOTA performance on PPI datasets, thereby underscoring the utility and efficacy of our proposed methodology.

**Graph Neural Networks**  GNNs represent a family of architectures that have demonstrated considerable success in various applications over recent years. More specifically, the majority of GNNs can be interpreted as a variant of the Message Passing Neural Networks (MPNNs) (Bronstein et al., 2021). From a technical standpoint, MPNNs operate based on an iterative paradigm. Upon initializing $\mathbf{h}_v^{(0)} = \mathbf{x}_v$ for each $v \in V$, where $\mathbf{x}_v$ denotes the feature vector of node $v$, MPNNs learn the embeddings $\mathbf{h}_v^{(l+1)}$ for each node $v \in V$ by executing the following steps during the $l$-th layer iteration:

$$\mathbf{h}_v^{(l+1)} = \phi^{(l)}\left(\mathbf{h}_v^{(l)}, \bigoplus_{u \in \mathcal{N}_v} \psi^{(l)}\left(\mathbf{h}_v^{(l)}, \mathbf{h}_u^{(l)}\right)\right).$$

In this context, $\mathbf{h}_v$ signifies its embedded feature. The superscripts serve to differentiate between layers, indicating the iteration number. The function $\psi$ represents a learnable message function that computes the message conveyed from node $u$ to $v$. The aggregation operation $\bigoplus$ collects the messages from the neighbors of node $v$, which then update the node features using the learnable function $\phi$. From the perspective of a single node, the core steps in the MPNN process involve the aggregation of messages from its neighboring nodes and the subsequent update of its feature by integrating its own characteristics with the aggregated message. Ultimately, the network outputs the node embedding $\mathbf{h}_v$ for every $v \in V$.

**PLH Integration**  Building upon earlier discussions, the output of the PLH is already vectorized for each node within a graph. This vectorization facilitates seamless integration of the PLH with various graph learning models. For instance, in the case of GNNs, existing initial node features $\mathbf{x}_v$ can be augmented by concatenation, thereby enriching the node's feature representation. Alternatively, the PLH output can be appended to the embedded feature $\mathbf{h}_v$ after iterations, enhancing the performance of subsequent processes in graph learning models.

**PLH-GNN Model**  PLH-GNN is a model designed to perform link prediction in a graph. The framework of the model is similar to TLC-GNN in Yan et al. (2021). The model extracts node features from both GNN and PLH. Then we transform the node features into edge features by applying operations such as elementwise sum, difference, squared difference, or just concatenation. The outputs of these transformations are denoted as $\mathbf{h}_{(u,v)}$ and $\mathbf{plh}_{(u,v)}$ respectively, representing the GNN-learnt and PLH-extracted features of the edge connecting nodes $u$ and $v$. By executing a concatenation operation, we generate $\mathbf{h}_{(u,v)} \bigoplus \mathbf{plh}_{(u,v)}$, which encapsulates information derived from both the GNN and PLH of our model. This enriched feature representation is subsequently fed into a MLP before the final prediction is made by the Fermi-Dirac Decoder (Krioukov et al., 2010; Nickel & Kiela, 2017; Yan et al., 2021).

## 5  Experiments

In the experiment section of our study, we comprehensively evaluate the performance enhanced by PLH in both synthetic and real-world scenarios, demonstrating their superiority over conventional PH. Specifically,

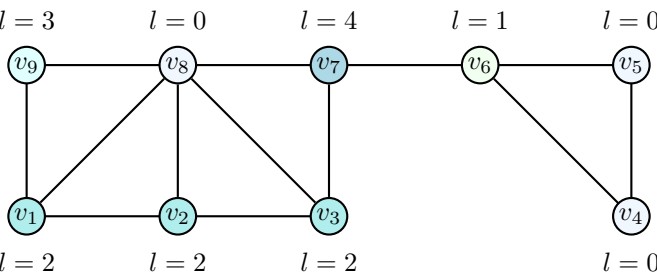

Figure 2: PLH-GNN Architecture. Node feature vector $\mathbf{h}_v$ is output by the GNN module, while $\mathbf{plh}_v$ is output by the PLH module. Edge feature $\mathbf{h}_{(u,v)}$ (or $\mathbf{plh}_{(u,v)}$ for PLH) is computed via a combination operation that merges the respective node features.

in the synthetic experiments, we meticulously design tasks to illustrate the enhanced capabilities of PLH compared to PH. Complementing this, our real-world experiments provide empirical evidence of PLH's overall superiority in practical applications. Furthermore, an ablation study reveals that integrating PLH into a SOTA model yields approximately a 1% increase in performance. The ablation study subsection also provides a more detailed comparative analysis of the computational efficiency between the rPLH and PLH algorithms. Notably, throughout our experiments, we employ various metrics to ensure a robust and fair comparison with existing baselines, reflecting the diverse tasks and datasets on which PLH is implemented. This multifaceted approach allows us to present a thorough evaluation of PLH's effectiveness across different contexts.

### 5.1 Synthetic Experiments

The synthetic experiments designed in this study aim to evaluate the effectiveness of the PLH in capturing local topological features within a graph, thereby augmenting the performance in graph-based learning tasks. Specifically, these experiments focus on the classification of nodes based on the count of 2-simplices (triangles) proximal to them, excluding their own presence. Considering an undirected graph $G = (V, E)$ where each edge denotes a unit distance, the assessment revolves around each node $v \in V$ and its punctured local subgraph $g_0^s(v)$. Here, $s$ represents the maximum predefined distance from the node $v$, and the label $l(v)$ quantifies the total number of 2-simplices within $g_0^s(v)$. For these experiments, we set $s = 2$, thus exploring a two-edge radius around each node. Figure 3 presents an illustrative example of this setup, providing a concrete visual representation of the classification of the nodes.

Figure 3: Illustration of a synthetic experiment: given $s = 2$, we focus on vertex $v_6$. The local graph $g_0^2(v_6)$ includes a single 2-simplex, specifically $[v_3, v_7, v_8]$, resulting in $l = 1$. It is important to note that $g_0^2(v_6)$ does not encompass $v_6$ itself, which implies that the simplex $[v_4, v_5, v_6]$ is not included.

We performed node classification tasks on three randomly generated graphs. The node features were adeptly represented using PH and PLH, with their filtration process contingent upon the dimension of simplices identified within each subgraph. To operationalize these features for classification, we employed the persistence image technique for vectorization. Subsequently, a Multi-Layer Perceptron (MLP) was utilized as the classifier. Hyperparameters remained consistent across all experimental setups and can be found in the Appendix.

The experimental results, as presented in Table 2, reveal a notable pattern: the accuracies achieved through the use of PLH consistently and significantly outperform those attained by PH across all cases. This disparity

Table 2: Experimental results on synthetic data. *Row C*: Represents the number of distinct classes in the dataset. *Row F*: Indicates the specific features utilized in the Multilayer Perceptron (MLP) model. *Comparison Metric*: The benchmark used is PH applied to local subgraphs denoted as $g^2(v)$. *Performance Measure*: The results are presented in terms of Accuracy, quantified as a percentage (%).

| C | 2 | | 3 | | 5 | | 7 | |
|---|---|---|---|---|---|---|---|---|
| F | PH | PLH | PH | PLH | PH | PLH | PH | PLH |
| $G_1$ | $93.63_{\pm2.67}$ | $\mathbf{100.00}_{\pm0.00}$ | $88.32_{\pm2.30}$ | $\mathbf{99.80}_{\pm0.69}$ | $81.38_{\pm3.86}$ | $\mathbf{95.30}_{\pm2.37}$ | $78.58_{\pm3.07}$ | $\mathbf{95.50}_{\pm3.55}$ |
| $G_2$ | $93.99_{\pm1.76}$ | $\mathbf{100.00}_{\pm0.00}$ | $91.49_{\pm1.40}$ | $\mathbf{99.69}_{\pm0.41}$ | $86.74_{\pm1.78}$ | $\mathbf{96.64}_{\pm1.15}$ | $84.57_{\pm3.69}$ | $\mathbf{93.57}_{\pm2.87}$ |
| $G_3$ | $96.15_{\pm0.90}$ | $\mathbf{99.96}_{\pm0.12}$ | $93.63_{\pm0.81}$ | $\mathbf{99.95}_{\pm0.10}$ | $90.31_{\pm1.06}$ | $\mathbf{98.53}_{\pm0.53}$ | $89.85_{\pm1.32}$ | $\mathbf{97.80}_{\pm1.09}$ |

in performance underscores the inherent advantage of PLH, particularly in its ability to capture topological information that remains elusive to PH. The results clearly demonstrate the unique strengths of PLH, providing empirical evidence of its superior capability in handling such tailored synthetic experiments.

## 5.2 Real-World Experiments

Although PLH demonstrated superior performance in synthetic experiments, the question arises regarding its efficacy in real-world applications. To address this, we conducted a series of experiments in real-world scenarios. These experiments were divided into two primary segments: OGB and PPI. In the OGB segment, we utilized datasets from Hu et al. (2020), integrating PLH into existing models without any model-specific fine-tuning. This approach allowed us to assess the adaptability and generalizability of PLH in a real-world context. For the PPI segment, we employed Protein-Protein Interaction (PPI) datasets, applying the PLH-GNN model as introduced in 4.4. Notably, the experiments conducted were limited to reduced PLH due to the complexity of real-world datasets.

### 5.2.1 Experiments on OGB Datasets

The primary objective of the experiments is to quantitatively assess the performance improvements achieved by the integration of PLH features, without the necessity for any model fine-tuning. For this purpose, we have employed benchmarks from the Open Graph Benchmark (OGB) (Hu et al., 2020) as our fundamental comparison baselines. Specifically, our experimental framework involved conducting node classification tasks on the ogbn-arxiv dataset, and link prediction tasks on the ogbl-ddi dataset. Detailed information regarding the datasets employed and the hyperparameters utilized in our experiments is comprehensively documented in the Appendix. The results of our experiments are systematically presented in Table 3a.

Table 3: Experimental results.

(a) Results for ogbn-arxiv and ogbl-ddi. Comparing with OGB baselines, we used identical hyperparameters in an ablation-style design to emphasize performance differences.

| Method | Accuracy (%) ogbn-arxiv | Hits@20 (%) ogbl-ddi |
|---|---|---|
| MLP | $55.50_{\pm0.23}$ | - |
| MLP w/ PLH | $\mathbf{57.74}_{\pm0.22}$ | - |
| GCN | $71.74_{\pm0.29}$ | $37.07_{\pm5.07}$ |
| GCN w/ PLH | $\mathbf{72.11}_{\pm0.29}$ | $\mathbf{45.23}_{\pm8.53}$ |
| SAGE | $71.49_{\pm0.27}$ | $53.90_{\pm4.74}$ |
| SAGE w/ PLH | $\mathbf{72.01}_{\pm0.38}$ | $\mathbf{63.73}_{\pm5.54}$ |

(b) Results on the PPI dataset. This table presents the outcomes from analyses conducted on the initial five graphs within the PPI dataset. Except for our results in the last row, all results are replicated from Yan et al. (2021).

| Method | AUROC (%) | | | | |
|---|---|---|---|---|---|
| | 1 | 2 | 3 | 4 | 5 |
| GCN | 75.21 | 74.42 | 77.68 | 76.22 | 69.67 |
| GIL | 57.69 | 1.45 | 34.90 | 85.61 | 33.65 |
| SEAL | 50.00 | 64.79 | 67.14 | 72.55 | 50.00 |
| TLC-GNN | 83.92 | 81.21 | 83.95 | 83.03 | 83.53 |
| PLH-GNN | $\mathbf{88.77}$ | $\mathbf{86.84}$ | $\mathbf{87.86}$ | $\mathbf{88.14}$ | $\mathbf{86.90}$ |

**Dataset ogbn-arxiv** A pivotal finding from our experiments with the ogbn-arxiv dataset is the notable performance enhancement of the Multi-Layer Perceptron (MLP) model when augmented with PLH features. This enhancement enables the augmented MLP model to surpass the performance of the traditional MLP model. The underlying reason for this improvement is primarily ascribed to the PLH's capability to leverage the topological structure of data. This ability enriches the model's predictive capabilities by providing additional contextual insights. In a similar vein, enhancements are also observable in GCN and GraphSAGE models when integrated with PLH. However, the extent of improvement in these models is relatively modest in comparison to that observed in the MLP model. This phenomenon is likely attributable to the intrinsic properties of GNNs, which are adept at extracting information from graph topological structures. Considering that GNNs already exhibit higher performance compared to MLP models, the marginal yet significant boost provided by PLH underscores its effectiveness.

**Dataset ogbl-ddi** In our investigation, link prediction tasks were performed on the ogbl-ddi dataset. To facilitate a direct comparison with the established baselines in the Open Graph Benchmark (OGB), we adapted our evaluation metric accordingly. The results were noteworthy: there was a substantial increase in performance, with the Hits@20 metric rising from 37.07% to 45.23% on the GCN. Furthermore, an impressive improvement in Hits@20 was observed, escalating from 53.90% to 63.73% on the GraphSAGE. These empirical findings strongly suggest that the integration of PLH significantly enhances the performance of both GCN and GraphSAGE models.

### 5.2.2 Experiments on PPI Datasets

The preceding experiments underscored the capability of PLH in augmenting MLP and GNNs through the integration of topological features derived from PLH. Building upon this foundation, the current experiments aim to explore the potential of PLH in enhancing a SOTA model. In this context, we applied the PLH-GNN model, as detailed in 4.4, to Protein-Protein Interaction (PPI) datasets for link prediction tasks Zitnik & Leskovec (2017). The model utilizes GraphSAGE for node embedding, capitalizing on its proficiency in encoding node features and graph structures effectively. To ensure a robust comparison with existing baselines, we employed the Area Under the Receiver Operating Characteristic (AUROC) as our primary evaluation metric. The comparative results are tabulated in Table 3b, juxtaposed against baselines including Graph Convolutional Network (GCN) (Kipf & Welling, 2016), Geometry Interaction Learning (GIL) (Zhu et al., 2020), SEAL (Zhang & Chen, 2018), and Topological Loop-counting Graph Neural Network (TLC-GNN) (Yan et al., 2021). The empirical evidence unequivocally demonstrates the superior performance of the PLH-GNN model, which achieves SOTA results on the PPI datasets, thereby attesting to its efficacy. This success prompted a subsequent ablation study to dissect the contributing factors of this achievement.

### 5.3 Ablation Study and Computational Efficiency

This study aims to address a lingering question from the previous section: to what extent does PLH contribute to SOTA performances? To investigate this, we conduct a comparative analysis of results obtained using PH, PLH, and their combined methodology (PH w/ PLH). These results are presented in Table 4.

The comparison is between the row representing the PLH, particularly the highlighted one identified as the top performer, and the baseline model, which is detailed in the first row of the table. This comparison indicates that integrating PLH into the GNN model results in an approximate 1% improvement over the existing SOTA performance. Notably, the baseline model itself is a SOTA model, and this slight yet significant enhancement further emphasizes the effectiveness of PLH in boosting the performance of the model.

A critical observation from the results presented in the table is the evident superiority of the combined method. Specifically, PH w/ PLH demonstrates a consistent performance edge over the standalone implementations of both PH and PLH. Remarkably, the inclusion of PLH enhances the model's capability, contributing to an additional performance increase of up to 0.53%. While this increment might seem marginal, it is important to note that this enhancement is realized by simply puncturing the center of local subgraphs. This outcome is noteworthy, as it represents a significant gain derived from a seemingly minimal modification. Furthermore, this finding corroborates our previous claims regarding the distinct capabilities of PLH in

Table 4: Ablation study for PPI in AUROC (%). $H$ is the maximum hop and $D$ is degree threshold. The first row is the results without any topological features, which services as the baseline.

| Feature | Parameters | 1 | 2 | 3 | 4 | 5 |
|---|---|---|---|---|---|---|
| - | - | $87.93_{\pm0.58}$ | $85.44_{\pm1.41}$ | $86.71_{\pm0.64}$ | $87.07_{\pm0.62}$ | $85.31_{\pm0.71}$ |
| PH | $H = 1, D = 4$ | $87.71_{\pm0.66}$ | $85.44_{\pm1.34}$ | $86.88_{\pm0.63}$ | $87.06_{\pm0.64}$ | $85.41_{\pm0.74}$ |
| PLH | $H = 1, D = 4$ | $87.67_{\pm0.53}$ | $85.67_{\pm0.82}$ | $86.70_{\pm0.65}$ | $87.09_{\pm0.68}$ | $85.35_{\pm0.75}$ |
| PH w/ PLH | $H = 1, D = 4$ | $\mathbf{87.87}_{\pm0.74}$ | $\mathbf{85.91}_{\pm0.94}$ | $\mathbf{87.11}_{\pm0.64}$ | $\mathbf{87.28}_{\pm0.63}$ | $\mathbf{85.94}_{\pm0.85}$ |
| PH | $H = 2, D = 4$ | $88.55_{\pm0.75}$ | $86.68_{\pm0.86}$ | $87.83_{\pm0.38}$ | $87.98_{\pm0.51}$ | $86.83_{\pm0.63}$ |
| PLH | $H = 2, D = 4$ | $88.49_{\pm0.68}$ | $86.66_{\pm0.82}$ | $87.82_{\pm0.45}$ | $88.00_{\pm0.47}$ | $86.67_{\pm0.73}$ |
| PH w/ PLH | $H = 2, D = 4$ | $\mathbf{88.65}_{\pm0.45}$ | $\mathbf{86.96}_{\pm0.84}$ | $\mathbf{88.01}_{\pm0.41}$ | $87.98_{\pm0.64}$ | $\mathbf{87.02}_{\pm0.52}$ |
| PH | $H = 2, D = 8$ | $88.81_{\pm0.59}$ | $86.98_{\pm0.87}$ | $87.90_{\pm0.57}$ | $88.11_{\pm0.55}$ | $86.80_{\pm0.60}$ |
| PLH | $H = 2, D = 8$ | $88.77_{\pm0.61}$ | $86.84_{\pm0.96}$ | $87.86_{\pm0.57}$ | $88.14_{\pm0.59}$ | $86.90_{\pm0.53}$ |
| PH w/ PLH | $H = 2, D = 8$ | $\mathbf{88.89}_{\pm0.41}$ | $\mathbf{87.12}_{\pm0.81}$ | $87.87_{\pm0.43}$ | $\mathbf{88.16}_{\pm0.45}$ | $\mathbf{86.96}_{\pm0.62}$ |
| PLH | $H = 2, D = \infty$ | $88.16_{\pm0.44}$ | $86.40_{\pm1.03}$ | $87.53_{\pm0.43}$ | $87.92_{\pm0.28}$ | $86.59_{\pm0.95}$ |

capturing topological characteristics that PH alone cannot. Consequently, this suggests that PH and PLH, as individual methods, provide unique and complementary topological insights.

**Comparing rPLH and PLH: Computational Efficiency** In the preceding discussion of real-world experiments, we did not explicitly distinguish between rPLH and PLH, treating rPLH as an alternate form of PLH. This approach is justified, as indicated by the final entry in Table 4, where rPLH effectively becomes identical to PLH when the threshold parameter $D$ is set to $\infty$. The computational times for both rPLH and PLH, calculated for various graphs, are presented in Table 5, with specific reference to the relevant data from Table 4. A critical observation is the significantly reduced computation time for rPLH compared to PLH in these datasets, underlining rPLH's scalability in practice.

Table 5: Comparison of computational time (in seconds) for rPLH and PLH on PPI datasets. For rPLH, values are rounded to the nearest unit, whereas for PLH (last line), they are rounded to the nearest hundred.

| Feature | Parameters | 1 | 2 | 3 | 4 | 5 |
|---|---|---|---|---|---|---|
| | $H = 1, D = 4$ | 2 | 2 | 4 | 4 | 2 |
| rPLH | $H = 2, D = 4$ | 4 | 3 | 5 | 6 | 4 |
| | $H = 2, D = 8$ | 6 | 5 | 9 | 10 | 6 |
| PLH | $H = 2, D = \infty$ | 500 | 500 | 3000 | 3500 | 700 |

## 6 Future Works and Conclusion

Our experiments confirm the effectiveness of PLH on smaller graphs; however, the scalability to larger graphs presents a considerable challenge due to the intricacies involved in processing simplicial complexes. More specifically, the computational complexity of PLH is denoted as $\mathcal{O}(|V|^{2\omega+1})$ when graphs are treated as 1-dimensional simplicial complexes. This represents a significant escalation from the $\mathcal{O}(|V|^{2\omega})$ complexity associated with PH on full graphs. Such an increase in computational demand notably constrains PLH's applicability for analysis of larger graphs. Although the reduced version, rPLH, enhances scalability by regulating the size and sparsity of the annular local subgraphs, it is imperative to maintain these subgraphs sufficiently close to their originals to effectively extract meaningful features. As such, the scalability still remains as an issue in practice. Consequently, future work will include developing scalable algorithms for simplicial complexes within graph data, aiming to enhance the applicability of PLH for comprehensive large-scale graph analysis.

While scalability remains a key challenge, there are two additional avenues for potentially improving our experimental results. The first approach focuses on enhancing the underlying prediction models used within PLH. The field of graph-related models is constantly evolving, and incorporating more advanced models such as RevGNN (Li et al., 2021) for node classification, or GNNs enhanced by the labeling trick (Zhang et al., 2021) and NCNC (Wang et al., 2023) for link prediction, could potentially lead to performance gains. Second, we can explore methods to directly improve the effectiveness of PLH itself. According to Hofer et al. (2020), making the filtration learnable can provide better expressive power, thereby potentially improving the performance of our method. Theoretical evidence supporting this idea can be found in a recent paper by Rieck (2023), which demonstrates that filtration plays a key role in the expressivity of PH in graph learning. Additionally, varying vectorization methods could present another avenue for improving the performance of PLH. A comprehensive survey by Ali et al. (2022) underscores the significant influence vectorization can have on performance. Considering these findings, investigating learnable filtrations and different vectorizations are promising strategies for enhancing the effectiveness of PLH.

In the context of our work, rPLH emerges as an intriguing alternative to PLH. Upon examination of Table 4, we observe that rPLH with proper parameter can surpass PLH. Various conjectures could explain this trend, such as noise and lack of topological properties due to high density in PLH; However, a more thorough investigation into the causal factors underlying the comparative efficiency of rPLH remains a promising area for future research, which would yield a deeper understanding of the mechanisms responsible for the observed performance enhancements.

Central to our study is the understanding that puncturing the central vertex aids in extracting more intricate topological information from local subgraphs. The procedure can be conceptualized as manipulating the filtration domain, allowing us to delve deeper into the topology of the data. As promising as this appears, it incites a compelling question: how can we harness the full extent of these latent topological properties? The task is nontrivial and demands meticulous exploration.

**Conclusion**   In summary, this study presents a mathematical formalization of Persistence Local Homology (PLH) for graphs, achieved by applying Persistence Homology (PH) to annular local subgraphs. We illustrate how PLH reveals local structures that are overlooked by conventional PH techniques applied to local subgraphs. Moreover, the benefits of this approach, which extend to graph learning methodologies incorporating PLH as features, are substantiated by our experiments demonstrating the efficacy of PLH in graph learnings. Significantly, the introduction of Reduced PLH enhances scalability while maintaining its effectiveness, underscoring its potential for practical applications in intricate graph-based learning environments.

### Acknowledgments

The research of Wang and Xu was supported in part by NSF through grant CCF-2200173 and by KAUST through grant CRG10-4663.2. Di Wang was supported in part by the baseline funding BAS/1/1689-01-01, funding from the CRG grand URF/1/4663-01-01, REI/1/5232-01-01, REI/1/5332-01-01, FCC/1/1976-49-01 from CBRC of King Abdullah University of Science and Technology (KAUST). He was also supported by the funding RGC/3/4816-09-01 of the SDAIA-KAUST Center of Excellence in Data Science and Artificial Intelligence (SDAIA-KAUST AI). The authors would like to express their sincere gratitude to the anonymous reviewers for their careful reading of the manuscript and their insightful comments and valuable suggestions.

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

## A More Related Works

**Graph Learning**  In this research, our primary objective is to investigate the effectiveness of PLH in the domain of graph learning tasks. Graph learning has recently gained significant attention due to its ability to model complex relationships within various types of data, leading to novel insights and improved performance in a multitude of applications Bronstein et al. (2021). The origins of graph learning can be traced back to graph theory and graph-based algorithms, which have been explored for several decades Easley & Kleinberg (2010). The emergence of contemporary machine learning approaches has accelerated advancements in this field, with deep learning on graphs emerging as an especially promising area of research. The amalgamation of these disciplines has given rise to a multitude of innovative algorithms and techniques, such as Graph Convolutional Networks (GCN) Kipf & Welling (2016), Graph Attention Networks (GAT) Veličković et al. (2017), and GraphSAGE Hamilton et al. (2017). Graph learning techniques have found applications in numerous domains, including social network analysis, recommendation systems, bioinformatics, and computer vision Zhou et al. (2020). For instance, in social networks, researchers have utilized graph learning methods to detect communities, link prediction, and identify influential nodes Cai et al. (2018). In the realm of bioinformatics, these methods have demonstrated potential in predicting protein-protein interactions, drug-target interactions, and gene-disease associations Zitnik et al. (2018). Additionally, graph learning has exhibited considerable potential in computer vision tasks, such as semantic segmentation and 3D shape analysis Gu et al. (2019).

**Node Classification and Link Prediction**  Graph learning, particularly in the realms of node classification and link prediction, has substantially propelled the progress in graph-based machine learning. In node classification tasks, the assignment of labels to graph nodes is informed by their attributes and the graph's structure. This field has evolved from heuristic methods to advanced machine learning techniques. For instance, early approaches, such as the one presented by Zhu & Ghahramani (2002), relied on the majority label of neighboring nodes for classification. The shift to more sophisticated models was marked by the integration of feature engineering and learning models, with significant contributions like Sen et al. (2008) combining structural and attribute information. The emergence of graph embedding techniques, including works by Perozzi et al. (2014) and Grover & Leskovec (2016), introduced the concept of representing nodes

as low-dimensional vectors. The most transformative development, however, has been the advent of Graph Neural Networks (GNNs), as evidenced by studies such as Kipf & Welling (2016), Veličković et al. (2017), Hamilton et al. (2017), and Chiang et al. (2019).

Link prediction, another significant graph task, is integral to analyzing complex networks with applications across various domains including social network analysis (Liben-Nowell & Kleinberg, 2003), recommendation systems (Koren et al., 2009; Adamic & Adar, 2003), biological networks (Coulomb et al., 2005), and knowledge graph completion (Teru et al., 2020). The goal here is to predict potential connections between pairs of nodes based on existing network data, which is fundamental in understanding network structures. Techniques for link prediction have evolved over time, ranging from similarity-based methods (e.g., common neighbors, Jaccard coefficient, Adamic-Adar) and matrix factorization to machine learning approaches like node2vec and graph convolutional networks (Grover & Leskovec, 2016; Kipf & Welling, 2016; Yan et al., 2021).

## B  Proof of Theorem 4.1

*Proof.* Noting that both subgraphs are finite, the numbers of both vertices and edges are finite. Hence the number of simplices in $\text{VR}_\epsilon\left(g_r^s\right)$ or $\text{VR}_\epsilon\left(g_r^{s\prime}\right)$ are finite. It shows that the number of births and deaths of topological features are finte. Consequently, there is a finite number of changes in the topological features as $\epsilon$ varies. Therefore, the persistence modules induced by the Vietoris-Rips filtration are q-tame. By Theorem 8 in Chazal & Michel (2021), we have $d_B\left(\text{dgm}_p, \text{dgm}_p'\right) \leq \max_{e \in \mathcal{E}} \mid w(e) - w'(e) \mid$. $\qquad\square$

## C  Fermi-Dirac Decoder

The Fermi-Dirac Decoder is a component of the PLH-GNN model and serves as a type of activation function. It is inspired by the Fermi-Dirac distribution from quantum mechanics, which describes the probability of finding particles in quantum states. In deep learning, what we take away mostly is the flexibility of the function. Here, we define the decoder as $F(u,v) = 1/(1 + e^{(d(u,v)-r)/t})$, where $d(u,v)$ is the metric between node $u$ and $v$, and $r,t$ are hyperparameters (Krioukov et al., 2010; Nickel & Kiela, 2017; Yan et al., 2021). It is worth noting that $F$ can be fine-tuned by the temperature parameter $t$: when $t$ is close to 0, the function becomes more step-like and introduces higher non-linearity, while when $t$ increases, the function becomes more linear and smooth. This flexibility can be useful in certain deep learning tasks where controlling the degree of non-linearity is desired.

## D  Experimental Details

### D.1  Experimental Environment

The experiments are conducted on a server, which is powered by an Intel(R) Xeon(R) W-2123 CPU clocked at 3.60GHz and is backed by 64 GB of RAM. The system is further enhanced with four GeForce RTX 2080 Ti GPUs, each with 11 GB of memory. On the software side, we utilize key packages for deep learning, with the main package versions being PyTorch 1.12.1, CUDA 11.3, and cuDNN 8. Additional dependency requirements are specified within the source code.

### D.2  Datasets

**Synthetic Datasets**   In the synthetic experiments, we utilize random graphs, commonly referred to as Erdős-Rényi graphs or binomial graphs (ERDdS & R&wi, 1959; Gilbert, 1959). These graphs are characterized by their edge selection process: for a given graph, each potential edge is independently chosen with a predefined probability denoted as $p$. The generation of the graphs was conducted using the hyperparameters specified in Table 6.

**Real-world Datasets**   Real-world datasets include ogbn-arxiv, ogbl-ddi, and Protein-Protein Interaction networks (PPI). Both the ogbn-arxiv and ogbl-ddi datasets are part of the Open Graph Benchmark (OGB) (Hu et al., 2020). OGB is an extensive collection of benchmark datasets, data loaders, and evaluators,

Table 6: Hyperparameters utilized for the generation of random graphs.

| Graph | #Nodes | $p$ |
|-------|--------|-----|
| $G_1$ | 1000 | 0.007 |
| $G_2$ | 2000 | 0.005 |
| $G_3$ | 5000 | 0.002 |

specifically designed for graph machine learning. The PPI dataset is derived from the research paper by Zitnik & Leskovec (2017). The PPI dataset comprises positional gene sets, motif gene sets, and immunological signatures. These various sets act as features, with a total count of 50. Additionally, the dataset includes gene ontology sets which serve as labels, numbered at 121 in total. For a summary of the fundamental characteristics of these datasets, we invite readers to consult Table 7. Further detailed information about these datasets is available in the original sources (Zitnik & Leskovec, 2017; Hu et al., 2020).

Table 7: Summary of real-world datasets.

| Dataset | #Graphs | #Nodes | #Edges | Task | Metric |
|---------|---------|--------|--------|------|--------|
| ogbn-arxiv | 1 | 169,343 | 1,166,243 | Node Classification | Accuracy |
| ogbl-ddi | 1 | 4,267 | 1,334,889 | Link Prediction | Hits@20 |
| PPI | 20 | 2,245.3 | 61,318.4 | Link Prediction | AUROC |

### D.3 Hyperparameters

**Filtrations** As discussed in Subsection 5.1, the filtration mechanism utilized in synthetic experiments is characterized by the dimension of simplices discerned within each subgraph. Precisely, the filtration function designates vertices with a value of 0, edges with 1, and 2-simplices with 2. This filtration approach is instrumental in identifying proximal triangles to a given vertex, primarily because edges and 2-simplices concurrently manifest in Vietoris-Rips filtrations when hop distance is employed. For empirical investigations in real-world scenarios, we employ Vietoris-Rips filtration, as introduced in Subsection 4.2.

**Experimental Hyperparameters** In this section, we delineate the hyperparameters critical for replicating the findings of our main study. Specifically, the symbol $H$ is utilized to represent the maximal hops within each local subgraph. Concurrently, $D$ signifies the threshold for degrees within the framework of the rPLH algorithm. Additionally, the resolution indicates the size of the persistent image.

Table 8: Hyperparameters. In synthetic experiments, the threshold parameter $D$ is set to $\infty$ adhering to the standard PLH algorithm's configuration.

| Experiment | $H$ | $D$ | Resolution | #Ephocs | #Runs |
|------------|-----|-----|------------|---------|-------|
| synthetic | 2 | $\infty$ | 8 | 100 | 20 |
| ogbn-arxiv | 2 | 8 | 16 | 500 | 10 |
| ogbl-ddi | 2 | 6 | 16 | 200 | 10 |
| PPI | 2 | 8 | 16 | 2500 | 20 |

