# OpenReview forum: "Persistent Local Homology in Graph Learning"
_TMLR — Accepted by TMLR_

### Review · Reviewer_gHvS · 2024-02-03

**Summary Of Contributions:**

This works proposes to use persistent local homology for graph learning. Persistent local homology is defined for the subgraph around each node within a radius d while the center node and its neighborhood within a radius r get removed. This work proposes this definition and also uses an example to show the more expressiveness of PLH than the traditional PH. The work also discusses who to simplify PLH by sub-selecting high-degree nodes. Some real world data experiments have been provided to verify the value of PLH.

**Audience:**

Yes

**Claims And Evidence:**

Yes

**Requested Changes:**

The requested changes are the answers to the weaknesses listed as above.

**Strengths And Weaknesses:**

Strengths:
1. The paper is very well written. The authors provide a clear position and sufficient background to understand the proposed PLH.
2. The more expressiveness of PLH is justified by a clear example.
3. Some practical problems such as PLH computation have been taken care empirically.
4. The experiments should some effectiveness of PLH.

Weaknesses:
1. The expressiveness of PLH is justified by just showing examples and compared with PH. Is it possible to add some quantitative justification of expressiveness, such as motif counting capability?  Also, how does it compare with WL tests?

2. The experiments are weak in this work.
For link prediction, can the authors use larger graphs from OGB for evaluation such as, PPA, COLLAB? The used DDI is small and dense, which often presents some properties different from common link prediction tasks. Also, when comparing methods over PPA, COLLAB, can the authors include the methods based on labeling tricks [1] as the baselines, such as SEAL. I noticed that the authors used SEAL for PPI but it is unfair to some extend because SEAL does not leverage node features, while for PPI,  node features are crucial. For PPA and COLLAB, as the node features here are not that important, the capability of capturing structure features, as what PLH aims to do, is more important. In particular, I do not think the current definition of PLH may address the node ambiguity issue as discussed in [1] for link prediction. I expect the authors may comment on this aspect. Also, whether PLH is scalable enough for these larger graphs is also a interesting question to investigate.

For node classification, the experiment is also weak, which just adopts one dataset. More datasets are suggested to be included.


[1] Labeling trick: A theory of using graph neural networks for multi-node representation learning, NeurIPS 2021, Zhang et al.

---

> ### Author Response · Authors · 2024-03-01
>
> We greatly appreciate the reviewer's thoughtful comments.
>
> **On Weaknesses:**
> ***
> *W1. On the expressiveness of PLH*
>
> The exploration of the expressiveness of PH remains a pivotal yet complex topic within the TDA community.
> The intricacies associated with quantitatively defining and theoretically guaranteeing this expressiveness extend beyond initial assumptions, indicating a rich field for future research.
>
> A recent advancement in the understanding of PH applied to graphs, as delineated by Bastian Rieck et al. [1], provides a foundational theorem that can be adapted to the context of PLH：
> > Theorem 5. Given $k$-FWL colourings of two graphs $G$ and $G^{\prime}$ that are different, there exists a filtration of $G$ and $G^{\prime}$ such that the corresponding persistence diagrams in dimension $k-1$ or dimension $k$ are different.
>
> Extending this theorem to PLH, we shall propose a similar theorem by considering annular local subgraphs in place of the entire graphs $G$ and $G^{\prime}$.
> However, the theorem's focus on local aspects does not directly address the global expressiveness of PLH, which is of paramount interest to our research community.
>
> The core inquiry thus becomes: Can PLH, through its analysis of annular local subgraphs, consistently differentiate between graphs $G$ and $G^{\prime}$ that are distinguishable by $k$-FWL colorings at a global scale?
> This question implicates the need for a comprehensive strategy to derive global descriptors from PLH representations and assess their impact on the overall expressiveness of the approach.
>
> Addressing this query requires us to navigate the transition from local to global graph descriptors via PLH, a process that poses significant theoretical and practical challenges.
> The development of such a global descriptor would not only augment the expressiveness of PLH but also enhance its applicability across a broader spectrum of graph-based data analysis tasks.
>
> In conclusion, while Theorem 5 offers a promising avenue for extending the theoretical underpinnings of PLH, significant work remains in exploring its global expressiveness capabilities.
> We believe this area of research holds the potential to substantially advance our understanding and utilization of TDA in graph learning, meriting further investigation and collaborative effort within the community.
>
>
> *W2. On the Experiments and Comparisons*
>
> * (On Experiments on Larger Graphs)
> The absence of experimental results on larger graphs, such as those from the Open Graph Benchmark (OGB), is a limitation we acknowledge. The computational demands of applying PLH to large-scale graphs have thus far hindered our ability to conduct these experiments efficiently. Our efforts to optimize computation, such as minimizing redundant PLH computations through change detection in graph dynamics, have yielded modest improvements. We recognize the need for more efficient approximation or implementation strategies to extend our analysis to larger graphs, a direction we are actively pursuing.
>
> * (On SEAL row in Table 3b) It's absolutely fine to remove the SEAL row from Table 3b since our main baseline is TLC-GNN.
>
> * (On the Node Ambiguity Issue) If I understand correctly, the node ambiguity issue arises because the method fails to distinguish between non-isomorphic links. This occurs because it treats node representations independently, without considering their interconnections.
> In light of this, your assertion seems correct; the PLH method indeed does not take into account the interconnections between the vertices under consideration. Consequently, it may not effectively address the node ambiguity issue.
> If the node ambiguity issue stems from the inability to differentiate nodes with isomorphic local subgraphs, then PLH potentially offers a solution through learnable filtration in supervised learning tasks. However, it's worth noting that this capability is not exclusive to PLH; learnable Persistent Homology (PH) also possesses the potential to address this challenge.
>
> We are grateful for the opportunity to clarify these aspects of our work and thank the reviewer for their constructive feedback.
>
> [1] Rieck, Bastian, et al. "On the Expressivity of Persistent Homology in Graph Learning."

---

> > ### Comment · Reviewer_gHvS · 2024-03-08
> > **Thanks for your response! Please adjust the manuscript by including the claimed limitations.**
> >
> > Thanks for your response! To be honest, I am not too satisfied with the response on larger graphs and comparing link prediction performance with labeling trick methods. If the authors are reluctant to address the concern, I suggest the authors at least revise the manuscript properly by incorporating some discussion on the limitations of this work.

---

> > > ### Author Response · Authors · 2024-03-09
> > >
> > > Thank you for your valuable feedback and constructive criticism.
> > >
> > > *Regarding Performance on Larger Graphs:*
> > >
> > > We acknowledge your concern regarding the scalability of PLH on larger graphs.
> > > In response, we have incorporated a detailed discussion in the manuscript to address this issue more clearly.
> > > Specifically, we added a paragraph to the beginning of Section 6, highlighting the scalability challenges of PLH and our current understanding of its limitations.
> > > Here is the revised paragraph:
> > >
> > > > Our experiments confirm the effectiveness of PLH on smaller graphs; however, the scalability to larger graphs presents a considerable challenge due to the intricacies involved in processing simplicial complexes.
> > > More specifically, the computational complexity of PLH is denoted as $\mathcal{O}(|V|^{2\omega+1})$ when graphs are treated as 1-dimensional simplicial complexes.
> > > This represents a significant escalation from the $\mathcal{O}(|V|^{2\omega})$ complexity associated with PH on full graphs.
> > > Such an increase in computational demand notably constrains PLH's applicability for analysis of larger graphs.
> > > Although the reduced version, rPLH, enhances scalability by regulating the size and sparsity of the annular local subgraphs, it is imperative to maintain these subgraphs sufficiently close to their originals to effectively extract meaningful features.
> > > As such, the scalability still remains as an issue in practice.
> > > Consequently, future work will include developing scalable algorithms for simplicial complexes within graph data, aiming to enhance the applicability of PLH for comprehensive large-scale graph analysis.
> > >
> > >
> > > In addition to the discussion above, we have explored practical approaches to mitigate the scalability issue, including:
> > > - Implementing parallel computation of PLH within a graph.
> > > - Dynamically computing PLH to avoid redundant calculations of similar annular local subgraphs.
> > >
> > > Despite these efforts, we recognize that these strategies do not fully overcome the scalability challenge.
> > > We believe that addressing this issue effectively requires more sophisticated methods and, potentially, better computational resources.
> > >
> > > *Comparing Performance with Labeling Trick Methods:*
> > >
> > >
> > > Regarding the comparison with methods using labeling tricks, we acknowledge that such a comparison would yield valuable insights. However, the feasibility of conducting this comparison hinges on overcoming the scalability issues we've mentioned previously. The main datasets featured in the studies of the labeling trick approach include ogbl-ppa, ogbl-collab, ogbl-ddi, and ogbl-citation2. Currently, our experiments cannot be expanded to cover ogbl-ppa, ogbl-collab, and ogbl-citation2 due to these scalability challenges. We considered limiting our comparison to the ogbl-ddi dataset alone. Yet, we foresee potential concerns from readers about the breadth and fairness of such a comparison. Nonetheless, we hold the view that labeling trick-based methods offer promising avenues for future research. This perspective has been incorporated into the discussion at the beginning of the second paragraph in Section 6.
> > >
> > > > While scalability remains a key challenge, there are two additional avenues for potentially improving our experimental results.
> > > The first approach focuses on enhancing the underlying prediction models used within PLH.
> > > The field of graph-related models is constantly evolving, and incorporating more advanced models such as RevGNN[1] for node classification, or GNNs enhanced by the labeling trick[2] and NCNC[3] for link prediction, could potentially lead to performance gains.
> > >
> > > We appreciate your suggestion to revise the manuscript to include a discussion on the limitations of our work.
> > > We hope that these revisions and clarifications address your concerns.
> > >
> > > [1] Training graph neural networks with 1000 layers.
> > > [2] Labeling trick: A theory of using graph neural networks for multi-node representation learning.
> > > [3] Neural common neighbor with completion for link prediction.

---

> > > > ### Comment · Reviewer_gHvS · 2024-03-17
> > > > **thanks for the response**
> > > >
> > > > Thank the authors for further improving the manuscript. I do not have more questions.

---

### Review · Reviewer_17y1 · 2024-02-12

**Summary Of Contributions:**

This work applies persistent local homology (PLH) to graph-structured data. The proposed method leverages Rips complexes built based on shortest-path distances in punctured local subgraphs. To alleviate the computational cost, the paper introduces a greedy pruning scheme to the local subgraphs based on node degrees--- the resulting method is called reduced PLH (rPLH). Vectorizations of the persistence diagrams are combined with GNN embeddings to obtain node- or edge-level predictions. Experiments on node classification and link prediction show the efficacy of the proposal.

**Audience:**

Yes

**Broader Impact Concerns:**

I have no concerns regarding broader impact.

**Claims And Evidence:**

Yes

**Requested Changes:**

Since I am inclined to accept the paper in its current form, I have only minor suggestions:

- Section 4.2: 'F as the selected filtration function' --- F is the filtration itself
- Section 4.1:  In the fragment "the punctured local subgraph ... is obtained by excluding the central vertex v from ..., where $0 < \epsilon < 1$ thus ...", it is not clear the role of epsilon here;
- Paragraph PLH-GNN Model: "we transform the node features into edge features by applying operations such as elementwise sum, difference, squared difference, or just concatenation" --- Since the paper only mentions undirected graphs, shouldn't this operation be order-invariant?
- Figure 2 is not clear, especially the path plh_v --> plh_{uv} (the same for h).
- In many applications, we are given graphs with edge features (or without them) instead of edge weights. Could you describe the Vietoris-Rips Filtration in this more general case?
- In Table 4, the baseline that doesn't use topological features achieves higher numbers than those from Table 3b. Could you explain why?

**Strengths And Weaknesses:**

**Strengths**
- The contribution is relatively novel, representing one of the first works applying PLH for graphs --- leveraging PH on annular local subgraphs.
- The paper is very well-written and easy to follow;
- The proposed method is simple and flexible --- it can be easily integrated into different GNNs.
- The paper reports experiments on artificial and real-world datasets and includes an ablation study.

**Weaknesses**
- The paper relies on a few examples to motivate the applicability of PLH with punctured local subgraphs. In this regard, can't we also create examples of graphs that are distinguishable via local subgraphs but not via the punctured ones? Overall, I found this motivation weak, and I think the paper would benefit from a theoretical analysis of the expressiveness of these approaches;

- The description of the rPLH method could be improved, for instance, by: i) discussing the motivation underlying the use of node degree as (node) importance measure; ii) clarifying why the computational analysis only relies on 0-dim and 1-dim simplices while the paper considers filtrations that result in higher-order simplicial complexes; iii) describing more precisely the pseudo-code --- e.g., using math notation instead of 'top-D i-th edge layer vertices' and indicating the variable r before using it). It would also be helpful to say that the subgraph induced by the sets of vertices (output of the procedure) defines the local subgraph used to obtain the filtration.

- Overall, the empirical gains do not seem statistically significant (see Table 4 -- PH vs PLH). Also, the experiments on real-world data (Table 3) do not report results for PH-based baselines.

---

> ### Author Response · Authors · 2024-03-01
>
> We greatly appreciate the reviewer's thoughtful comments.
> ***
> **On Weaknesses:**
>
> *W1. On the expressiveness of PLH:*
> We appreciate the reviewer's insight on the expressiveness of PLH.
> Kindly check the fist part of the reply to reviewer gHvS.
>
> *W2. Regarding rPLH:*
> * (i) We acknowledge the reviewer's concern about the heuristic nature of rPLH. The design of rPLH aims to preserve as many topological features as possible while simplifying the graph structure. High-degree nodes are considered of paramount importance due to their extensive connectivity, suggesting a richer preservation of topological characteristics during the reduction process. It is a heristic method.
> * (ii) The complexity analysis primarily focuses on 0-dimensional and 1-dimensional simplices by treating graphs as 1-simplices, which aligns with the most of scenarios in existing literature. If we consider all high-dimensional simplices, the number of simplices goes to $2^{|V|} -1$, which would make the complexity comparesion less evident.
> * (iii) The description of pseudocode of rPLH has been enhanced with a clearer comment.
>
> *W3. Experimental Gains:*
> We acknowledge the reviewer's observation regarding the empirical gains presented in Table 4 and Table 3's real-world data experiments.
> While the improvements with PLH over PH are modest, they illustrate PLH's potential utility, as discussed in the first part of our response to reviewer XuSs.
> ***
> **On Requested Changes:**
>
> * We have corrected the terminology in Section 4.2 to accurately refer to 'F' as the filtration function, rather than the filtration itself.
> * The role of $\varepsilon$ in the definition of the punctured local subgraph has been clarified by renaming the subgraph to $g_0^s$ in Section 4.1, enhancing the readability and understanding of its function.
> * In the link prediction of undirected graphs, it is usually preferable to use order-invariant edge features for better generalization. We initially used concatenation in our PPI experiments because it slightly outperformed other methods in the initial probing tests. We speculate that the additional size of the neural network gained through concatenation compensates for this deficiency.
> * Figure 2's clarity has been improved with a detailed caption, ensuring a better understanding of the relationships depicted within the model architecture.
> * For graphs incorporating edge features, these features can be transformed into edge weights through methods such as function mapping, normalization, or neural network-based techniques. This adaptation enables the use of Vietoris-Rips Filtration in broader contexts. In cases where an edge lacks features or weights, assigning a unit weight to the edge is a viable approach.
> * The discrepancy in Table 4's baseline performance is attributed to the application of different subgraph extraction methods and the GraphSAGE model. We hypothesize that GraphSAGE's architecture significantly contributes to the observed performance gains, offering a plausible explanation for the comparative results.
>
> We are grateful for the opportunity to clarify these aspects of our work and thank the reviewer for their constructive feedback.

---

### Review · Reviewer_XuSs · 2024-02-16

**Summary Of Contributions:**

In this article, the authors propose to use local persistent homology for machine learning on graphs. More precisely, they show that, instead of considering the Vietoris-Rips persistence diagrams of subgraphs (as it is currently done in the literature on local persistent homology), one can rather look at punctured and annular subgraphs, that is, subgraphs made of the vertices which are at distance at least r and at most s from a given vertex v (r,s are parameters). In fact, these subgraphs can be seen as generalizations of the link of v, whose reduced homology is the same as the (mathematical) local homology group of v, that is, the homology group of the whole space X relative to X without v. Motivated by this connection, the authors provide a stability result for the persistent homology of these local annular subgraphs, and demonstrate their usefulness by applying it on several node classification tasks, where they show positive comparison of their approach over both persistent homology on ordinary subgraphs and over non-topological baselines (on synthetic data), or positive comparison of their approach combined with persistent homology on ordinary subgraphs (on real-world graph data).

**Audience:**

Yes

**Claims And Evidence:**

No

**Requested Changes:**

---Would it be possible to prove an approximation quality result for rPLH? Something like an upper bound (that depends on D) for the bottleneck distance between the persistence diagrams of regular and annular subgraphs? Typically, this upper bound would decrease as D increases. I think this would further motivate the use for rPLH, in addition to better running times.

---I would recommend to introduce punctured subgraphs as special cases of annular subgraphs where the r parameter is between 0 and 1. Right now, it is a bit weird to see the notation with epsilon for punctured subgraphs, as it is not clear before seeing the notation of annular subgraphs, which appear later in the text.

---It would be good to be explicit about the r and s parameters that are used to compute the annular subgraphs in Figure 1.

---It would be nice to also provide the bandwidth parameters used for the persistence images in Section D.3, as this parameter is also very important.

**Strengths And Weaknesses:**

This article is very well written and easy to follow, and the application of topological descriptors to graph data is an important one in machine learning. My main concern however is that I struggle to understand the actual added value of looking at punctured and annular subgraphs rather than simply taking the persistent homology of regular subgraphs. I understand Figure 1 and Table 1, but I think one could make a symmetric argument: it is easy to build examples where regular local subgraphs are different (in terms of homology groups), but annular local subgraphs are the same. For instance, take the following graphs (described with their edges): (v_0, v_1), (v_0, v_2), (v_0, v_3), (v_1, v_2), (v_2, v_3), (v_0, v_4), (v_0, v_5), (v_0, v_6), (v_4, v_5), (v_5, v_6) and (v_0, v_1), (v_0, v_3), (v_1, v_2), (v_2, v_3), (v_0, v_4), (v_0, v_6), (v_4, v_5), (v_5, v_6).

Then the annular subgraphs (around the v_0 vertices) with parameters r=1 and s=2 are the same (betti 0 = 2, betti 1 = 0), while the regular subgraphs are different (betti 1 = 4 (up) and 2 (bottom)). My criticism also extends to the experiments: as far as I understand, on the synthetic data, the goal is to predict the structures of the annular subgraphs around the vertices, so it is not surprising that the persistent homology of annular subgraphs work better than the persistent homology of regular subgraphs, it feels like a circular (empirical) demonstration. On actual data, in particular Table 4, it is not obvious that annular subgraphs work systematically better than regular ones, sometimes they do, sometimes not, but results are overall quite similar. Hence, given that the stability result is just an application of a known theorem, I have difficulties to see where the contribution is. It would be nice to provide data sets where the prediction goal is oblivious to the types of subgraphs, and on which annular subgraphs are more efficient than regular ones.

---

> ### Author Response · Authors · 2024-03-01
>
> We greatly appreciate the reviewer's thoughtful comments.
> ***
> **On Weaknesses:**
>
> *W1. The added value of examining PH on punctured and annular subgraphs over regular subgraphs:*
>
> It is indeed possible to construct scenarios where the PH of regular local subgraphs differ while the PH of annular local subgraphs remain identical, as the reviewer points out. The core of our contribution lies in demonstrating that PLH can capture nuances in topological feature that PH may overlook. The degree to which PLH enriches our understanding varies with the dataset; our findings show a pronounced effect in synthetic datasets and a more subtle impact in the context of PPI datasets. We understand that the similarities in performance between PLH and PH on PPI may raise concerns about the added value of PLH. Allow me to draw an analogy to human vision to elucidate the potential benefit: the slight difference in perspective between our two eyes allows us to perceive depth, enriching our two-dimensional sight with a three-dimensional understanding of the world. While we do not claim that PLH will universally augment PH to a similar extent, our preliminary results suggest its potential to reveal additional structural insights in certain datasets.
>
> *W2. Do the synthetic experiments like a circular demonstration?*
>
> We appreciate the reviewer's insightful comments regarding our experimental design, particularly the use of synthetic and real-world data. In response to the concerns raised about the synthetic experiments seeming circular, we wish to clarify that these experiments were intentionally designed to highlight the distinct advantages of persistent homology of annular local subgraphs (PLH) over traditional persistent homology (PH) on local subgraphs. Our primary objective is not to assert that PLH universally outperforms PH, but rather to demonstrate that PLH offers unique benefits in specific contexts (PLH $\neq$ PH). In this case, it supports our argument that PLH can extract additional features that are beneficial for machine learning prediction tasks, thus validating our computation pipeline. Regarding the mixed results observed in Table 4, we acknowledge that the superiority of annular subgraphs over regular ones is not consistently evident across all datasets.
> ***
> **On Requested Changes:**
>
> We acknowledge the reviewer's constructive suggestions and have addressed each point as follows:
>
> * In relation to the quality of the approximation for rPLH, we agree with the proposal to establish an upper bound on the bottleneck distance between the persistence diagrams of regular and reduced subgraphs, based on $D$. We acknowledge the intricacy of this endeavor, given its reliance on multiple parameters such as the inner radius $r$, outer radius $s$, and $D$, as well as the filtration process. Further exploration in this direction calls for additional efforts.
>
> * We have revised the manuscript to introduce punctured subgraphs as special cases of annular subgraphs to avoid ambiguity.
>
> * The parameters $r$ and $s$ utilized in the computation of annular subgraphs for Figure 1 have been explicitly detailed in the revised manuscript.
>
> * Regarding the bandwidth parameters for persistence images in Section D.3, we wish to clarify that the bandwidth was not manually specified in our experiments. Instead, we employed Persim's adaptive or data-driven approach to bandwidth selection. This method ensures that the persistence images are both informative and robust by optimizing the bandwidth based on the underlying data distribution.
>
> We are thankful for the opportunity to improve our work in response to the reviewer's feedback.

---

> > ### Comment · Reviewer_XuSs · 2024-03-04
> >
> > Thank you for the clarification.

---

### Decision · Action_Editor_mZBK · 2024-03-19

**Recommendation:** Accept as is

**Comment:**

All reviewers found the paper interesting and  well written but had  few comments and questions. The authors addressed those comments and updated the paper to reflect them. The reviewers acknowledged and appreciated those discussions and changes that made the paper stronger.

Despite one reviewer that found the contribution to be limited,  all reviewers agree that the claims have been demonstrated theoretically and empirically. Since the paper is technically sound and the results presented in this paper are of interest to the community, the paper fits all acceptance criteria and I recommend acceptance.

**Audience:**

This paper with contribution in the field of graph neural network  is definitely of interest to the community of TMLR.

**Claims And Evidence:**

The paper propose a novel way to integrate local persistent homology  and provides a numerically efficient greedy scheme to limit complexity. All claims in the paper (stability, complexity, performance gain in some cases) have been supported as acknowledged by all reviewers.